# Polyporenic Acids from the Mushroom *Buglossoporus quercinus* Possess Chemosensitizing and Efflux Pump Inhibitory Activities on Colo 320 Adenocarcinoma Cells

**DOI:** 10.3390/jof9090923

**Published:** 2023-09-13

**Authors:** Kristóf Felegyi, Zsófia Garádi, Bálint Rácz, Gábor Tóth, Viktor Papp, Imre Boldizsár, András Dancsó, Gabriella Spengler, Szabolcs Béni, Attila Ványolós

**Affiliations:** 1Department of Pharmacognosy, Semmelweis University, 1085 Budapest, Hungary; felegyi.kristof@pharma.semmelweis-univ.hu (K.F.); garadi.zsofia@pharma.semmelweis-univ.hu (Z.G.); boldizsar.imre@semmelweis.hu (I.B.); beni.szabolcs@semmelweis.hu (S.B.); 2Directorate of Drug Substance Development, Egis Pharmaceuticals Plc., 1475 Budapest, Hungary; dancso.andras@egis.hu; 3Albert Szent-Györgyi Health Center, Department of Medical Microbiology, Albert Szent-Györgyi Medical School, University of Szeged, 6725 Szeged, Hungary; racz.balint@med.u-szeged.hu (B.R.); toth.gabor.1@med.u-szeged.hu (G.T.); spengler.gabriella@med.u-szeged.hu (G.S.); 4ELKH-USZ Biologically Active Natural Products Research Group, University of Szeged, 6720 Szeged, Hungary; 5Department of Botany, Hungarian University of Agriculture and Life Sciences, 1118 Budapest, Hungary; papp.viktor@uni-mate.hu; 6Department of Plant Anatomy, Institute of Biology, Eötvös Loránd University, 1117 Budapest, Hungary; 7Department of Analytical Chemistry, Institute of Chemistry, Eötvös Loránd University, 1117 Budapest, Hungary

**Keywords:** polyporenic acid, chemosensitizing, efflux pump inhibitory, Colo 320

## Abstract

Polyporenic acids N-R (**1**–**5**), five novel 24-methylene lanostane triterpenes along with seven known polyporenic acids (**6**–**12**), were identified from the fruiting bodies of *Buglossoporus quercinus*. The isolation of compounds **1**–**12** was performed by a combination of multistep flash chromatography and reversed-phase high-performance liquid chromatography (HPLC). The structure determination was carried out by extensive spectroscopic analysis, including 1D and 2D nuclear magnetic resonance (NMR) and high-resolution electrospray ionization mass spectrometry (HR-ESI-MS) experiments. The isolated fungal metabolites were investigated for their antiproliferative activity in vitro by 3-(4,5-dimethylthiazol2-yl)-2,5-diphenyltetrazolium bromide (MTT) assay on the resistant Colo 320 human colon adenocarcinoma cell line expressing P-glycoprotein (ABCB1). The lanostane triterpenes exerted moderate antiproliferative activity with IC_50_ values in the range of 20.7–106.2 μM. A P-glycoprotein efflux pump modulatory test on resistant Colo 320 cells highlighted that fungal metabolites **3**, **5**, **8**, and **10**–**12** have the ability to inhibit the efflux pump activity of cancer cells. Moreover, the drug interactions of triterpenes with doxorubicin were studied by the checkerboard method. Compounds **3**–**4**, and **7**–**12** interacted in a synergistic manner, while an outstanding potency was detected for compound **9**, which was defined as strong synergism (CI = 0.276). The current study reveals that *B. quercinus* is a remarkable source of fungal steroids with considerable chemosensitizing activity on cancer cells.

## 1. Introduction

Fungi, as members of the third-largest kingdom on Earth, are considered prolific synthesizers of natural products of an astonishing structural diversity including triterpenes, cyclic peptides, polysaccharides, alkaloids, and ceramides, among others. Compared to plant-derived compounds, fungal secondary metabolites were previously considered a somewhat neglected area of natural products. However, in the last decade, the research performed on fungal metabolites has clearly been intensified. Special attention has been focused on the large group of triterpenes as one of the most characteristic secondary metabolites of fungal organisms. Members of the order *Polyporales*, the so-called polypore fungi, have a great contribution to the overall number of triterpenes identified from fungal species. Recent studies indisputably demonstrate their ability to produce triterpenes with notable structural diversity and significant biological activity, for example, irpexolidal, a compound with an unprecedented carbon skeleton from the medicinal fungus *Irpex lacteus* [1]; antimalarial tomophagusins from *Tomophagus* sp. [2]; and lanostane triterpenes with antidiabetic potential from *Fomitopsis pinicola* [3], to mention only a few recent studies. Among the more than 100 genera of the vast order of *Polyporales*, there is a lesser-known genus, called *Buglossoporus*, which was established by Kotlába and Pouzar in 1966 to accommodate *B. quercinus* (Schrad.) Kotl. and Pouzar, a rare brown-rot polypore species described from Europe [4]. This species was also discussed in the genus *Piptoporus* P. Karst. [5], since its basidiomata shows morphological similarity to the birch polypore, *P. betulinus* (Bull.) P. Karst. (its current name is *Fomitopsis betulina*), a well-studied medicinal mushroom from Europe [6]. However, phylogenetic studies showed that the two species are not closely related, and the *Buglossoporus* genus has an independent systematic position within the antrodia clade [7,8]. In addition to *B. quercinus*, only two non-European and little-known *Buglossoporus* species are confirmed by phylogenetic evidence: *B. americanus* D.A. Reid has only been found in Costa Rica [9], while *B. eucalypticola* M.L. Han, B.K. Cui, and Y.C. Dai is known from the Tropical Botanical Garden in Hainan Province, Southern China [7]. In Europe, *B. quercinus* is widely distributed but considered to be a rare and threatened species associated with old oak trees [10]. The basidiomata of *B. quercinus* is annual, pileate, thick- and pale-flashed with a yellow-brown surface and whitish pore surface with roundish pores. The recognition of this species in the field is further facilitated by the vinaceous discoloration of fresh basidiomata when bruised, the perfumed smell of the flesh, and its distinctly bitter taste.

Although many species in the antrodia clade are known as significant and well-studied medicinal mushrooms (e.g., *Fomitopsis betulina*, *F. pinicola*, *Laetiporus sulphureus*, *Laricifomes officinalis*, and *Wolfiporia hoelen*) [6], according to our literature review there is no available study on the chemical composition and biological activity of any *Buglossoporus* species. Therefore, we aimed to examine the chemical profile of fruiting bodies of *B. quercinus* to isolate the most important secondary metabolites and determine the potential pharmacological activities.

## 2. Results and Discussion

### 2.1. Structure Determination of Compounds

In-depth chemical experiments were performed to unveil the characteristic secondary metabolites of the brown-rot fungus, *B. quercinus*. The collected mushroom material was freeze-dried and then extracted with methanol at room temperature. The crude extract was subjected to solvent–solvent partition with *n*-hexane, chloroform, and ethyl acetate. The chloroform extract was separated in multiple steps of normal and reversed-phase flash chromatography. The final purification performed by reversed-phase HPLC led to the identification of compounds **1**–**12** (Figure 1).

The structure determination was carried out by extensive spectroscopic analysis, including 1D and 2D NMR and HR-ESI-MS experiments. Based on the HR-ESI-MS measurement, the molecular formula of compound **1** is C_31_H_48_O_5_. The NMR spectra of the sample suggested a 24-methylene lanostane skeleton with the characteristic resonances of the methylene moiety at C-31 (*δ*_C_ 106.9) and H-31 (*δ*_H_ 4.61). The resonances at *δ*_C_ 136.6 and 134.0 indicated the presence of an unsaturation in the triterpenoid backbone. According to their HMBC correlations with methyl groups H_3_-30 and H_3_-19, they were assigned to C-8 and C-9, respectively. The resonance at *δ*_H_ 3.89 with HSQC correlation to *δ*_C_ 77.1 allowed us to conclude that the lanostane skeleton also carries a hydroxyl group. Its position at C-16 was confirmed by COSY cross-peaks with adjacent atoms and NOE correlations to H_3_-19 and H-20. *δ*_C_ 177.4 and its HMBC correlations to H-20 and H-17 revealed a carboxylic group at the C-20 position. The compound also contained a keto group (*δ*_C_ 216.0) in position 3, assigned by the HMBC correlations to H_2_-1, H_2_-2, and H_3_-29. Considering the lanostane backbone, a rather unusual hydroxymethyl resonance appeared at *δ*_H_ 3.39 and 3.16 with HSQC cross-peaks to *δ*_C_ 68.3. HMBC correlations of the diastereotopic protons with C-3 and C-5 suggested C-28 or C-29 as the site of the hydroxymethyl group. Their NOE correlation with H-5 confirmed the hydroxymethyl group in position 28 (shown in Figure 2). Thus, compound **1** was assigned as 16,28-dihydroxy-24-methylene-3-oxo-lanost-8-en-21-oic acid and named polyporenic acid N.

Related 29-hydroxy lanostane derivatives were identified in *Poria cocos* [11], while the 4,4-dimethyl derivative of polyporenic acid N was reported from *Daedaleopsis confragosa* var. *tricolor* [12].

According to HR-ESI-MS data, the molecular formula of compound **2** was found to be C_31_H_48_O_4_. The compound exhibited the same molecular formula as polyporenic acid H [13] (compound **9**); however, significant ^1^H and ^13^C NMR resonance differences were observed in positions 3, 4, 5, 28, and 29 (see Table 1). While in the case of compound **9**, NOE correlations were detected between H-3 and H_3_-19, H_3_-28, and H_3_-29 (shown in Figure 2), in the case of compound **2**, NOE correlations were shown between H-3 and H-5, H_3_-28 (can be seen in Figure 2).

These correlations highlighted that compounds **9** and **2** differ in the configuration of the hydroxyl group at the C-3 position. To provide further evidence, the ^1^H NMR resonances and the H-3 multiplicity also differed from that of the 3α-OH substitution pattern [14]. Contrary to 3α-OH, the 3β-OH substitution is characterized by an indicative H-3–H-5 NOE correlation and the ~dd multiplicity of the H-3 resonance (see Figure 2 and Appendix A). Thus, compound **2** was found to be the C-3 epimer of polyporenic acid H (**9**), determined as 3β-hydroxy-24-methylene-12-oxo-lanost-8-en-26-oic acid and named polyporenic acid O.

HR-ESI-MS data indicated a molecular formula of C_33_H_50_O_5_ for compound **3**. The NMR data of **3** were similar to that of polyporenic acid H [13] (compound **9**) except for resonances of an extra acetyl group. According to the HMBC correlations between the ester carbonyl (δ_C_ 170.8) and H-3, the polyporenic acid H skeleton bears the acetyl ester group at the C-3 position. The coupling pattern [14] and the observed NOE correlations with only the “adjacent” (H_2_-2, H_3_-28, and H_3_-29) protons of the H-3 proton suggested the α-orientation of the substituent. Therefore, compound **3** was assigned as 3α-*O*-acetylpolyporenic acid H and named polyporenic acid P.

Based on the HR-ESI-MS data, a molecular formula of C_34_H_52_O_6_ for compound **4** can be presented. The 1D and 2D NMR spectra of the compound suggested a 24-methylene lanostane backbone by the typical methylene resonances at C-24 (*δ*_C_ 148.3), exhibiting a ^13^C chemical shift at *δ*_C_ 111.4 and ^1^H chemical shifts at *δ*_H_ 4.97, 4.94 at C-31. A carboxylic group (*δ*_C_ 179.6) was assigned to the C-25 position by their HMBC correlations with H-25 and H_3_-27. The compound contained an unsaturation (*δ*_C_ 134.3 and 134.2), whose position at C-8–C-9 was established by their HMBC cross-peaks C-8/H_3_-30 and C-9/H_3_-19. This polyporenic acid structure (named polyporenic acid D, C_31_H_50_O_3_) was described by Thappa et al. in 1981 [15], yet no detailed NMR characterization of this compound has been published since then. Moreover, the elemental composition difference and the HMBC correlations of H_2_-2′/C-1′, C-3′ of **4** indicated the presence of an additional malonyl moiety to the previously described skeleton. The position of the substituent was confirmed by the HMBC correlation between H-3 and C-1′ (malonyl carbonyl group). Since the NOE correlations and multiplicity pattern of the H-3 proton were characteristic of a 3α-O substitution [14], compound **4** was described as 3α-malonyl-24-methylene-lanost-8-en-26-oic acid (3α-malonylpolyporenic acid D) and named polyporenic acid Q.

Data on HR-ESI-MS of compound **5** showed a molecular formula of C_38_H_60_O_8_. The NMR spectra of **5** also proposed a 24-methylene lanostane backbone. An unsaturation between C-8–C-9 and a carbonyl group attached to C-25 was confirmed by their HMBC cross-peaks. The resonance at *δ*_H_ 4.27 (d, *J* = 8.0 Hz) with HSQC correlation to *δ*_C_ 72.5 suggested the presence of an oxymethine moiety. Based on their COSY cross-peaks with H_2_-11 and the HMBC correlations to C-9, C-14, and C-18, a hydroxyl group was assigned to the C-12 position. The key correlations of H-12/H-18/H-20 confirmed the α-orientation of 12-OH. Furthermore, Zhao et al. previously reported that the doublet multiplicity of the H-12 resonance suggests an α-orientation, while the triplet-like multiplicity indicates a β-orientation of 12-OH in the ^1^H NMR spectra of lanostane-type C_31_ triterpenoids [16]. Although this described polyporenic acid structure (named polyporenic acid A, C_31_H_50_O_4_) was isolated and reported in several previous works, no reliable NMR assignment can be found for polyporenic acid A (hereby referred to as compound **10**) [17,18,19,20]. In addition, the NMR spectra showed a singlet resonance at *δ*_H_ 3.67 with an HSQC cross-peak at *δ*_C_ 52.2. This characteristic resonance with the HMBC correlation to C-26 indicated that compound **5** is a methyl-ester derivative of polyporenic acid A at C-26. MS/MS fragmentation also suggested the presence of a 3′-hydroxy-3′-methyl-glutaryl moiety, however, the NMR spectra recorded in pyridine-*d*_5_ did not support this assumption. To avoid signal broadening, the NMR spectra of **5** were also recorded in a methanol-*d*:pyridine-*d*_5_ 19:1 solvent mixture. By using this solvent mixture, the ^1^H and ^13^C resonances of the proposed substituent and their 2D NMR correlations could be observed. Therefore, this novel polyporenic A derivative was assigned as methyl-3α-(3′-hydroxy-3′-methyl-glutaryl)-24-methylene-12α-hydroxy-lanost-8-en-26-oate, and named polyporenic acid R. Regarding the absolute configuration of polyporenic acids, in a previous study written by King et al. [18], the absolute configuration of C_25_ was established based on single crystal X-ray crystallographic analysis of the methyl ester of polyporenic acid A, and it was found to be 25 (S). Another study by Kamo et al. [18] presented the results of a semisynthetic preparative work performed on derivatives of polyporenic acid A to determine the absolute configuration. The results confirmed that the examined fungal metabolites are (25S,3′S)-(+)-12R-hydroxy-3R-(3′-hydroxy-3′-methylglutaryloxy)-24-methyllanosta derivatives. Assuming that the same metabolic pathways lead to the recently isolated polyporenic acid N-S and the previous members of the same series of polyporenic acids, an identical configuration of chiral centers is suggested for compounds described in the current study.

The known compounds were identified as (25S)-(+)-12R-hydroxy-3R-malonyloxy-24-methyllanosta-8,24(31)-dien-26-oic acid [18] (compound **6**), (25S,3′S)-(+)-12α-hydroxy-3α-[3′-hydroxy-3′-methylglutaryloxy]-24-methyllanosta-8,24(31)-dien-26-oic acid [17,20] (compound **7**), (25S)-(+)-12α-hydroxy-3α-methylcarboxyacetate-24-methyllanosta-8,24(31)-diene-26-oic acid [21,22] (compound **8**), polyporenic acid H [13] (compound **9**), polyporenic acid A [18] (compound **10**), 3α-*O*-acetylpolyporenic acid A [16,21], (compound **11**), and polyporenic acid C [13,18,21], (compound **12**) based on their NMR spectra, HRMS data and previously reported characteristics.

### 2.2. Antiproliferative Activity

The isolated compounds (**1**−**12**) were tested for their antiproliferative activity on resistant Colo 320 cell line using the 3-(4,5-dimethylthiazol2-yl)-2,5-diphenyltetrazolium bromide (MTT) assay with doxorubicin used as a positive control. The examined fungal constituents demonstrated antiproliferative activity against the tumor cell line with IC_50_ values in the range of 20.7–106.2 μM (Table 2). Compounds **3**, **11**, and **12** exerted a more pronounced activity, while **1**–**2** and **6** proved to possess a weaker inhibitory activity.

### 2.3. MDR Efflux Pump Inhibitory Activity

The isolated fungal metabolites **1**–**12** were further investigated for their potential efflux pump inhibitory activity by measuring the intracellular accumulation of rhodamine 123, a widely used P-glycoprotein (ABCB1, P-gp) substrate fluorescent dye, within the MDR Colo 320 cells expressing P-gp. Tariquidar, a potent P-gp inhibitor, was applied as the positive control. The examined constituents were dissolved in DMSO, and then the final concentration (2.00%) of the solvent was investigated for any effect on the retention of rhodamine 123. Typically, the fungal metabolites with fluorescence activity ratio (FAR) values greater than 2 were considered to be active P-gp inhibitors, whereas the isolates with FAR values higher than 10 were considered to be strong MDR modulators. According to our results, notable inhibitions of P-gp MDR efflux pump activity were revealed for compounds **3**, **5**, **8**, and **10**–**12**. Among the tested fungal metabolites, polyporenic acid R (**5**) bearing a 3′-hydroxy-3′-methyl-glutaryl moiety at C_3_ proved to be the most active with a 7.611 FAR value (Table 3).

### 2.4. Combination Studies with Doxorubicin

Compounds **3–4**, and **7**–**12** were selected for the study of their capacity to decrease the resistance of the MDR Colo 320 cell line to doxorubicin. A checkerboard microplate combination assay was carried out, which is one of the most suitable in vitro methods for the evaluation of drug interactions. The obtained data were analyzed using CompuSyn software ver. 1.0 (ComboSyn Inc., Paramus, NJ, USA), which allowed the identification of the most efficient ratios of combined agents and the calculation of combination indices (CI). According to the combination indices, the type of interaction could be defined for the examined combinations of fungal metabolites and doxorubicin. Compounds **3**–**4**, and **7**–**12** behaved in a synergistic way with CI values at 50% of the ED50, which were found to be 0.601, 0.691, 0.419, 0.574, and 0.608, respectively (Table 4). Remarkable potency was detected for compound **9**, which was defined as strong synergism (CI = 0.276). Interestingly, compound **9** exerted moderate efflux pump inhibitory activity while demonstrating strong synergism in combination with doxorubicin, which raises questions about the possible mechanism behind the observed pharmacological activity of compound **9**.

## 3. Materials and Methods

### 3.1. General Experimental Procedures

The optical rotations were determined using a Jasco P-2000 digital polarimeter (JASCO International, Co., Ltd., Hachioji, Tokyo, Japan) at the Na_D_ line.

The structures were determined by high-resolution mass-spectrometry methods. The Dionex Ultimate 3000 UHPLC system (3000RS diode array detector, TCC-3000RS column thermostat, HPG-3400RS pump, SRD-3400 solvent rack degasser, and WPS-3000TRS autosampler) (Thermo Fischer Scientific, Waltham, MA, USA) was used connected to an Orbitrap Q Exactive Focus Mass Spectrometer with an electrospray ionization source (Thermo Fischer Scientific, Waltham, MA, USA). The ionization source was operated both in positive and negative ionization modes, and operation parameter optimization was automatic using the built-in software. The following working parameters were applied: spray voltage (+), 3500 V, spray voltage (−), 2500 V; capillary temperature, 320 °C; sheath gas (N2), 47.5 °C; auxiliary gas (N2), 11.25; pare gas (N2), 2.25 arbitrary units. The full scan resolution was 70,000, and the scanning range was between 120 and 2000 *m*/*z* units. Parent ions were fragmented with a normalized collision energy of 15%, 30%, and 45%. The VDIA isolation range selection was selected by previous measurements. The samples were prepared in methanol and filtered through MF-Millipore membrane filters (0.45 μm, mixed cellulose esters) (Millipore Sigma, Billerica, MA, USA).

NMR spectra were recorded in deuterated chloroform (chloroform-*d*, 99.8 atom% D, with 0.03% (*v*/*v*) TMS, Sigma-Aldrich Kft., Budapest, Hungary), methanol (methanol-*d*_4_, 99.8 atom% D, with 0.03% (*v*/*v*) TMS, Sigma-Aldrich), pyridine (pyridine-*d*_5_, 99.8 atom% D, Sigma-Aldrich), or tetrahydrofuran (tetrahydrofuran-*d*_8_, 99.5 atom% D, Sigma-Aldrich) on a Bruker Avance III HD 600 (600/150 MHz) instrument equipped with a Prodigy cryo-probehead (Bruker Biospin GmbH, Rheinstetten, Germany) at 295 K. The pulse programs were taken from the Bruker software library (TopSpin 3.5, pl 7). ^1^H and ^13^C chemical shifts (*δ*) are given in ppm relative to the NMR solvent or relative to the internal standard (TMS), while the coupling constants (*J*) are given in Hz. The complete ^1^H and ^13^C assignments (shown in Appendix A) were achieved by widely accepted strategies based on ^1^H NMR, ^13^C NMR, ^1^H-^1^H COSY, ^1^H-^1^H TOCSY, ^1^H-^1^H NOESY, ^1^H-^13^C HSQC, and ^1^H-^13^C HMBC correlations.

Flash chromatography (FC) was performed on a CombiFlash Rf+ Lumen instrument with integrated UV and UV–Vis detections using normal-phase (silica 20, 40, and 80 g, 0.045–0.063 mm, Molar Chemicals, and RediSep Rf Gold C18, Teledyne Isco, Lincoln, NE, USA) and reversed-phase flash columns (30, 50, and 150 g RediSep Rf Gold C18, Teledyne Isco, Lincoln, NE, USA). Reversed-phase HPLC separations were carried out on a Waters 2690 HPLC system equipped with a Waters 996 diode array detector (Waters Corporation, Milford, MA, USA). As the stationary phase, a Kinetex C18 100 Å (150 × 10 mm i.d., 5 µm; Phenomenex Inc., Torrance, CA, Canada) column was used. The chemicals used were provided by Sigma-Aldrich Kft. (Budapest, Hungary) and Molar Chemicals (Halásztelek, Hungary).

### 3.2. Mushroom Material

Sporocarps of *B. quercinus* were collected in Vértes Mountains, Hungary (47.379678, 18.327107) on 6 August 2020, and authenticated by one of the authors (Viktor Papp). The fungal samples were cleaned from any pollution, i.e., soil contaminants or plant parts, and then stored at −20° C. A voucher specimen (No. VPapp-2008061) has been deposited at the Department of Botany, Hungarian University of Agriculture and Life Sciences, Budapest, Hungary.

### 3.3. Extraction and Isolation

The fresh mushroom material (2.1 kg) was lyophilized to obtain a 205 g dried sample, then percolated with MeOH (10 L) at room temperature. The dry methanol extract (45 g) was dissolved in 50% aqueous MeOH (600 mL) and solvent−solvent partition was performed using *n*-hexane (3 × 300 mL), CHCl_3_ (3 × 300 mL), and then EtOAc (3 × 300 mL). The chloroform fraction (20 g) was further purified with normal-phase flash chromatography (NP-FC) using a gradient system of *n*-hexane and acetone. Following separation, fractions with similar compositions were combined according to thin-layer chromatography (TLC) monitoring (BQ1–BQ15). The selected fraction of BQ6 (44 mg) was further separated by a combination of normal and reversed-phased flash chromatography (RP-FC) and finally purified by reversed-phase HPLC (mobile phase: H_2_O-MeOH, 85 to 95% MeOH gradient elution) to give compound **3** (1.5 mg). Combined fraction BQ7 (200 mg) was first separated by RP-FC (mobile phase: H_2_O-MeOH), then purified by RP-HPLC (mobile phase of water: methanol, 75–85% MeOH gradient elution), leading to compound **10** (2.4 mg). Compounds **2** (2 mg), **4** (3.8 mg), **5** (1.9 mg), and **9** (1.2 mg) were isolated from fraction BQ8 (616 mg) by a combination of RP-FC (mobile phase of methanol-water) and RP-HPLC (mobile phase of water: methanol, 75 to 100% MeOH gradient elution)**.** Fraction BQ10 (5.12 g) was further purified in consecutive steps of normal-phase FC using eluent systems of *n*-hexane: acetone and chloroform: methanol affording compounds **8** (900 mg) and **11** (2.7 mg). Compounds **1** (0.9 mg) **7** (0.8 mg), and **12** (400 mg) were isolated from fraction BQ11 (6.53 g) upon a combination of NP-FC (mobile phase of *n*-hexane-acetone) and RP-FC (mobile phase of water: methanol), followed by a RP-HPLC purification (mobile phase of water: methanol, 80 to 95% MeOH gradient elution). Compound **6** (90 mg) was obtained after a separation of BQ12 (310 mg) performed on RP-FC (mobile phase of water: methanol) followed by a final purification on RP-HPLC (mobile phase of water: methanol, 75 to 85% MeOH gradient elution).

Polyporenic acid N (**1**): amorphous solid; $\alpha_{D}^{25}$ + 10.4 (*c* 0.3, CH_3_OH); HR-ESI-MS (+) *m/z* 501.3568 [M + H]^+^ (501.3575 calcd for C_31_H_49_O_5_ Δ −1.3 ppm); ^1^H and ^13^C NMR data, see Table 1; HR-ESI-MSMS (CID = 15%, 30%, 45%) 483.3460, 465.3365, 453.3355.

Polyporenic acid O (**2**): amorphous solid; $\alpha_{D}^{25}$ + 26.7 (*c* 0.2, CH_3_OH); HR-ESI-MS (+) *m/z* 485.3620 [M + H]^+^ (485.3625 calcd. for C_31_H_49_O_4_ Δ −1.0 ppm); ^1^H and ^13^C NMR data, see Table 1; HR-ESI-MSMS (CID = 15%, 30%, 45%) 468.9432, 454.8377.

Polyporenic acid P (**3**): amorphous solid; $\alpha_{D}^{25}$+1.1 (*c* 4.4, CH_3_OH); HR-ESI-MS (+) *m/z* 527.3721 [M + H]^+^ (527.3731 calcd. for C_33_H_51_O_5_ Δ −2.0 ppm); ^1^H and ^13^C NMR data, see Table 1; HR-ESI-MSMS (CID = 15%, 30%, 45%) 509.3611, 481.3671, 467.3504, 449.3400, 421.3456.

Polyporenic acid Q (**4**): amorphous solid; $\alpha_{D}^{25}$+5.5 (*c* 0.4, CH_3_OH); HR-ESI-MS (−) *m/z* 555.3694 [M − H]^−^ (555.3680 calcd. for C_34_H_51_O_6_ Δ 2.6 ppm); ^1^H and ^13^C NMR data, see Table 1; HR-ESI-MSMS (CID = 15%, 30%, 45%) m/z 511.3787.

Polyporenic acid R (**5**): amorphous solid; $\alpha_{D}^{25}\;$− 2.6 (*c* 4.3, CH_3_OH); HR-ESI-MS (+) *m/z* 645.4342 [M + H]^+^ (645.4361 calcd. for C_38_H_61_O_8_ Δ 2.8 ppm); ^1^H and ^13^C NMR data, see Table 1; HR-ESI-MSMS (CID = 15%, 30%, 45%) 627.4243, 465.3715.

### 3.4. Cell Culture

The human colon adenocarcinoma cell lines, Colo 205 (ATCC-CCL-222) doxorubicin-sensitive and Colo 320/MDR-LRP (ATCC-CCL-220.1) resistant to doxorubicin expressing P-glycoprotein (ABCB1), were purchased from LGC Promochem (Teddington, UK). The cells were cultured in RPMI-1640 medium supplemented with 10% heat-inactivated fetal bovine serum (FBS), 2 mM L-glutamine, 1 mM Na-pyruvate, 10 mM Hepes, nystatin, and a penicillin−streptomycin mixture in concentrations of 100 U/L and 10 mg/L, respectively. The MRC-5 (ATCC-CCL-171) human embryonic lung fibroblast cell line (LGC Promochem) was cultured in EMEM medium, supplemented with 1% nonessential amino acid mixture, 10% heat-inactivated FBS, 2 mM L-glutamine, 1 mM Na-pyruvate, nystatin, and a penicillin−streptomycin mixture in concentrations of 100 U/L and 10 mg/L, respectively. The cell lines were incubated in a humidified atmosphere (5% CO_2_, 95% air) at 37 °C.

### 3.5. Assay for Antiproliferative Effect

The effects of increasing concentrations of the compounds on cell growth were evaluated in 96-well flat-bottomed microtiter plates. The 2-fold serial dilutions of the isolated constituents were made starting with 100 μM. Then, 6 × 103 human colonic adenocarcinoma cells in 100 μL of the medium (RPMI-1640) were added to each well, except for the medium control wells. Culture plates were incubated at 37 °C for 72 h; at the end of the incubation period, 20 μL of MTT (thiazolyl blue tetrazolium bromide) solution (from a 5 mg/mL stock solution) was added to each well. After incubation at 37 °C for 4 h, 100 μL of sodium dodecyl sulfate (SDS) solution (10% SDS in 0.01 M HCl) was added to each well, and the plates were further incubated at 37 °C overnight. Cell growth was determined by measuring the optical density (OD) at 540 nm (ref 630 nm) with a Multiscan EX ELISA reader (Thermo Labsystems, Cheshire, WA, USA). Inhibition of cell growth was expressed as IC_50_ values defined as the inhibitory dose that reduces the growth of the cells exposed to the tested compounds by 50%. IC_50_ values and the SD of triplicate experiments were calculated by using GraphPad Prism software version 5.00 for Windows with nonlinear regression curve fit (GraphPad Software, ver. 9.0.1, San Diego, CA, USA; www.graphpad.com, accessed on 15 March 2022).

### 3.6. Rhodamine 123 Accumulation Assay

The cell numbers of the human colon adenocarcinoma cell lines were adjusted to 2 × 106 cells/mL, resuspended in serum-free RPMI-1640 medium, and distributed in 0.5 mL aliquots into Eppendorf centrifuge tubes. The analyzed fungal metabolites were added at 20 μM concentrations, and the samples were incubated for 10 min at room temperature. Tariquidar was used as the positive control at 0.2 μM. DMSO at 2% *v/v* was applied as solvent control. Then, 10 μL (5.2 μM final concentration) of the fluorochrome and ABCB1 substrate rhodamine 123 (Sigma-Aldrich) were added to the samples, and the cells were incubated for a further 20 min at 37 °C, washed twice, and resuspended in 1 mL of PBS for analysis. The fluorescence of the cell population was determined with a PartecCyFlow flow cytometer (Partec, Münster, Germany). The FAR was calculated as the quotient between FL-1 of the treated/untreated resistant Colo 320 cell line over the treated/untreated sensitive Colo 205 cell line according to the following equation:$$\mathrm{FAR}\;=\frac{\mathrm{Colo}320_{\mathrm{treated}}/\mathrm{Colo}320_{\mathrm{control}}}{\mathrm{Colo}205_{\mathrm{treated}}/\mathrm{Colo}205_{\mathrm{control}}}$$

### 3.7. Checkerboard Combination Assay

In the present study, a checkerboard microplate method was used to investigate the effect of drug interactions between the compounds and the chemotherapeutic drug doxorubicin. The assay was performed on the Colo 320 colon adenocarcinoma cell line. The final concentration of the compounds and doxorubicin used in the combination experiment was chosen in accordance with their antiproliferative activity towards this cell line. The dilutions of doxorubicin were made in a horizontal direction in 100 μL, and the dilutions of the compounds vertically in the microtiter plate in 50 μL volume. Then, 6 × 10^3^ of Colo 320 cells in 50 μL of the medium were added to each well, except for the medium control wells. The plates were incubated for 72 h at 37 °C in a 5% CO_2_ atmosphere. The cell growth rate was determined after MTT staining. At the end of the incubation period, 20 μL of MTT solution (from a stock solution of 5 mg/mL) was added to each well. After incubation at 37 °C for 4 h, 100 μL of SDS solution (10% in 0.01 M HCI) was added to each well and the plates were further incubated at 37 °C overnight. OD was measured at 540 nm (ref. 630 nm) with a Multiscan EX ELISA reader. Combination index (CI) values at 50% of the growth inhibition dose (ED_50_) were established using CompuSyn software (ComboSyn, Inc., Paramus, NJ, USA) to plot four to five data points at each ratio. CI values were calculated by means of the median-effect equation, according to the Chou–Talalay method, where CI < 1, CI = 1, and CI > 1 represent synergism, additive effect (or no interaction), and antagonism, respectively [23,24].

## 4. Conclusions

The current study represents the first chemical and pharmacological investigation of *B. quercinus*, a lesser-known brown-rot polypore, the single European member of the *Buglossoporus* genus. Comprehensive chemical analysis of the methanol extract obtained from the freeze-dried sporocarps of *B. quercinus* resulted in the identification of five new natural products, named polyporenic acids N-R (**1**–**5**), and seven known polyporenic acid derivatives (**6**–**12**). The pharmacological experiments performed on resistant Colo 320 cells revealed that fungal triterpenoids **3**, **5**, **8**, and **10**–**12** have the ability to inhibit the efflux pump activity of cancer cells. Drug interaction experiments of triterpenes with doxorubicin highlighted that compounds **3**–**4**, and **7**–**12** interact in a synergistic manner, while a remarkable potency was revealed for compound **9**, which was defined as strong synergism. Our thorough report on *B. quercinus* undoubtedly demonstrates that this polypore species is rich in bioactive natural products, which warrants further studies for its potential valorization as a medicinal mushroom.

## Figures and Tables

**Figure 1 jof-09-00923-f001:**
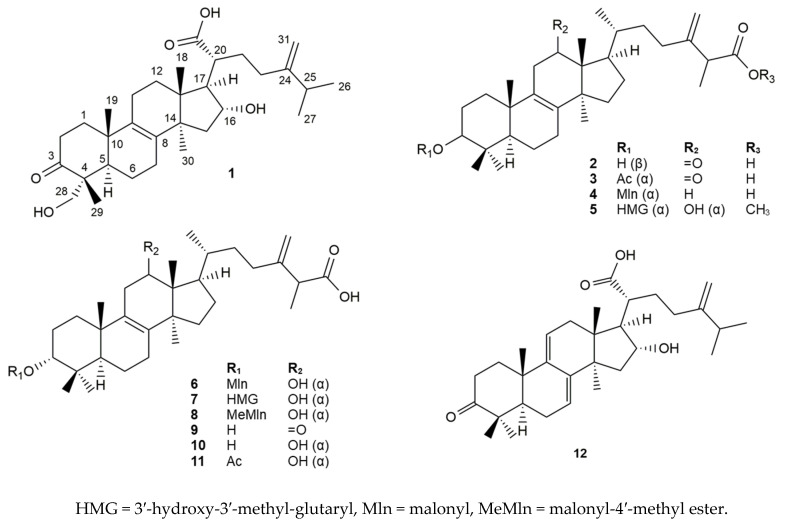
Compounds **1**–**12** isolated from *Buglossoporus quercinus.*

**Figure 2 jof-09-00923-f002:**
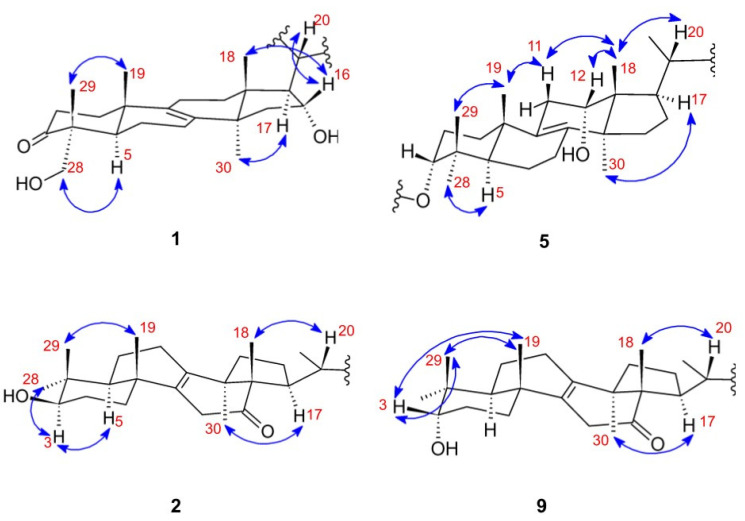
Key NOESY correlations of polyporenic acid N (**1**), polyporenic acid R (**5**), polyporenic acid O (**2**), and polyporenic acid H (**9**).

**Table 1 jof-09-00923-t001:** NMR spectroscopic data (600 MHz, CDCl_3_
^a^, pyridine-*d*_5_
^b^, tetrahydrofuran-*d*_8_
^c^, or methanol-*d*:pyridine-*d*_5_ 19:1 ^d^) for compounds **1**–**5.**

	1 ^c^	2 ^c^	3 ^a^	4 ^a^	5 ^b^
Position	δ_C_ Type	δ_H_ (*J* in Hz)	δ_C_ Type	δ_H_ (*J* in Hz)	δ_C_ Type	δ_H_ (*J* in Hz)	δ_C_ Type		δc Type	δ_H_ (*J* in Hz)
1	35.7	CH_2_	1.82, m	36.4	CH_2_	1.51, m	30.7	CH_2_	1.43, m	30.6	CH_2_	1.49, m	31.6	CH_2_	1.75, m
1.54, m	1.09, m	1.37, m	1.41, m	1.55, m
2	36.8	CH_2_	2.31, dd, (10.0, 8.0)	28.8	CH_2_	1.48, m	23.1	CH_2_	1.87, m	23.2	CH_2_	1.90, m	24.1	CH_2_	1.90, m
2.28, dd, (8.0, 3.4)	1.67, m	1.72, m	1.78, m
3	216.0	C		78.3	CH	2.97, m	77.6	CH	4.68, br s	80.6	CH	4.76, br s	78.4	CH	4.97, br s
4	52.8	C		39.7	C		36.7	C		36.8	C		37.4	C	
5	44.3	CH	2.17, m	51.5	CH	0.96, m	45.2	CH	1.50, m	45.3	CH	1.47, m	46.4	CH	1.83, m
6	20.2	CH_2_	1.49, m	19.1	CH_2_	1.65, m	17.8	CH_2_	1.67, m	18.0	CH_2_	1.59, m	18.8	CH_2_	1.62, m
1.44, m	1.49, m	1.56, m	1.49, m	1.52, m
7	27.2	CH_2_	2.03, m	27.2	CH_2_	2.06, m	25.8	CH_2_	2.12, m	25.9	CH_2_	2.04, m	26.8	CH_2_	2.17, m
1.93, m	1.95, m	2.07, m	2.12, m
8	136.6	C		135.8	C		135.0	C		134.2	C		135.5	C	
9	134.0	C		136.6	C		135.1	C		134.3	C		134.2	C	
10	37.6	C		37.8	C		36.7	C		36.8	C		37.5	C	
11	21.4	CH_2_	1.94, m	40.5	CH_2_	2.82, m	39.9	CH_2_	2.90, m	20.9	CH_2_	2.01, m	34.9	CH_2_	2.76, m
2.49, m	2.71, m	2.50, m
12	30.0	CH_2_	1.71, m	210.5	C		212.7	C		30.9	CH_2_	1.74, m	72.5	CH	4.27, d (7.6)
1.39, m	1.69, m
13	46.5	C		55.0	C		54.5	C		44.5	C		50.5	C	
14	49.4	C		60.0	C		59.4	C		49.9	C		50.4	C	
15	44.0	CH_2_	2.03, m	30.9	CH_2_	1.75, m	30.1	CH_2_	1.82, m	30.8	CH_2_	1.60, m	33.1	CH_2_	1.76, m
1.15, m	1.20, m	1.35, m	1.18, m	1.25, m
16	77.1	CH	3.89, m	28.6	CH_2_	1.91, m	27.8	CH_2_	2.01, m	28.2	CH_2_	1.94, m	28.7	CH_2_	2.11, m
1.29, m	1.37, m	1.31, m	1.40, m
17	57.3	CH	1.96, m	43.5	CH	2.06, m	42.4	CH	2.19, m	50.3	CH	1.50, m	43.5	CH	2.65, m
18	17.8	CH_3_	0.69, s	12.9	CH_3_	0.94, s	12.6	CH_3_	1.05, s	15.7	CH_3_	0.69, s	17.2	CH_3_	0.77, s
19	19.2	CH_3_	0.95, s	19.5	CH_3_	0.98, s	19.0	CH_3_	1.08, s	18.9	CH_3_	1.00, s	19.4	CH_3_	1.03, s
20	48.3	CH	2.24, m	37.7	CH	1.18, m	36.6	CH	1.27, m	36.3	CH	1.41, m	37.3	CH	1.53, m
21	177.4	C		19.6	CH_3_	0.78, d (6.4)	19.0	CH_3_	0.89, d (5.8)	18.6	CH_3_	0.93, m	18.3	CH_3_	1.28, m
22	31.6	CH_2_	1.92, m	35.3	CH_2_	1.53, m	34.0	CH_2_	1.62, m	34.2	CH_2_	1.58, m	35.4	CH_2_	1.77, m
1.64, m	1.14, m	1.27, m	1.19, m	1.38, m
23	33.4	CH_2_	1.96, m	32.9	CH_2_	2.14, m	32.1	CH_2_	2.21	31.7	CH_2_	2.19, m	32.7	CH_2_	2.37, m
1.87, m	1.09, m	2.02	2.00, m	2.17, m
24	156.7	C		150.9	C		148.5	C		148.3	C		150.2	C	
25	34.8	CH	2.14, m	46.2	CH	2.96, m	45.1	CH	3.17, m	45.0	CH	3.18, q (7.0)	46.2	CH	3.32, q (7.1)
26	22.4	CH_3_	0.91, d (6.9)	175.6	C		177.7	C		179.6	C		175.3	C	
27	22.2	CH_3_	0.92, d (6.9)	17.0	CH_3_	1.10, d (7.4)	16.3	CH_3_	1.31, d (6.7)	16.1	CH_3_	1.31, d (7.0)	17.1	CH_3_	1.38, d (7.1)
28	68.3	CH_2_	3.39, dd (10.2, 2.4)	28.5	CH_3_	0.88, s	27.5	CH_3_	0.88, s	27.6	CH_3_	0.88, s	28.5	CH_3_	1.04, s
3.16, dd (10.2, 2.4)
29	17.5	CH_3_	0.78, s	16.1	CH_3_	0.69, s	21.7	CH_3_	0.94, s	21.7	CH_3_	0.93, m	22.4	CH_3_	0.91, s
30	25.3	CH_3_	1.05, s	24.3	CH_3_	0.65, s	24.1	CH_3_	0.82, s	24.2	CH_3_	0.91, s	25.7	CH_3_	1.43, s
31	106.9	CH_2_	4.61, br s	110.3	CH_2_	4.77, br s	111.1	CH_2_	4.97, br s	111.4	CH_2_	4.97, br s	111.3	CH_2_	5.10, br s
4.71, br s	4.94, br s	4.94, br s	5.06, br s
1′					170.8	C		167.2	C		172.4 ^d^	C	
2′							40.2	CH_2_	3.64, m	47.0 ^d^	CH_2_	2.64 ^d^, m
3′							170.6	C		71.0 ^d^	C	
4′										47.0 ^d^	CH_2_	2.64 ^d^, m
5′										172.4 ^d^	C	
1′-CH_3_					21.3	CH_3_	2.07, s				
3′-CH_3_										27.9 ^d^	CH_3_
26-CH_3_										52.3 ^d^	CH_3_

**Table 2 jof-09-00923-t002:** Antiproliferative activity of compounds **1**–**12**.

	Colo 320 (IC_50_ µmol)	
Compound	Mean	SD
1	85.65	1.45
2	106.20	7.14
3	29.78	2.77
4	69.16	7.47
5	36.55	0.91
6	87.37	3.29
7	61.71	3.86
8	36.18	2.73
9	39.46	1.71
10	48.97	1.24
11	29.74	0.36
12	20.71	2.13
doxorubicin *	0.39	0.09

* positive control.

**Table 3 jof-09-00923-t003:** P-gp efflux pump inhibitory activity of compounds **1**−**12** on the MDR Colo 320 colon adenocarcinoma cell line.

Compound	conc µmol	FAR
**1**	20	0.774
**2**	20	1.051
**3**	20	4.448
**4**	20	1.338
**5**	20	7.611
**6**	20	0.386
**7**	20	0.593
**8**	20	2.079
**9**	20	0.942
**10**	20	3.822
**11**	20	2.491
**12**	20	2.267
Tariquidar ^a^	0.2	15.091
DMSO	2%	0.402

^a^ positive control.

**Table 4 jof-09-00923-t004:** Chemosensitizing activity of compounds **3**–**4** and **7**–**12** on Colo 320 adenocarcinoma cells.

Compound	Best Ratio	CI at ED_50_	SD	Interaction
**3**	27.6:1	0.601	0.049	synergism
**4**	64.1:1	0.691	0.070	synergism
**7**	114.5:1	0.419	0.069	synergism
**8**	67.2:1	0.574	0.069	synergism
**9**	586.2:1	0.276	0.096	strong synergism
**10**	90.9:1	0.779	0.082	moderate synergism
**11**	27.6:1	0.841	0.140	moderate synergism
**12**	38.4:1	0.608	0.032	synergism

## Data Availability

Not applicable.

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
