# Peer review of "Polyporenic Acids from the Mushroom Buglossoporus quercinus Possess Chemosensitizing and Efflux Pump Inhibitory Activities on Colo 320 Adenocarcinoma Cells"

_jof, 2023, doi:10.3390/jof9090923_

Round 1
Reviewer 1 Report
The manuscript submitted by Kristóf Felegyi et al. reported a series of compounds polyporenic acids with chemosensitizing and efflux pump inhibitory activities, discovered from the fruiting bodies of Buglossoporus quercinus. In the process, they performed by combination of multistep flash chromatography and reversed phase HPLC. The structure determination was carried out by extensive spectroscopic analysis, including 1D and 2D NMR and HRESIMS experiments. Overall, above work provided some insights into polypore species is rich in bioactive natural products.
The detail comments are listed as follows.
1. The manuscript just described the relative configuration of partial chiral center, however the absolute configuration of all five novel compounds is not confirmed. The author should confirm the absolute configuration by ECD or single crystal diffraction.
2. The number of the subtitle should be re-checked, especially the order of “3. Results and Discussion” and “2. Materials and Methods” (latter).
3. The more details of determination of compound structure should be described, for example, in line 138, “NOE correlations of the H-3 proton”; line 153, “the NOE” and line 161, “Based on their 2D NMR correlations”
4. The table 1, please check the format of the three-wire meter.
5. Figure 1, the configuration of C-20 should be presented.
The English writing is fine.
Author Response
Thank you for the helpful and valuable comments you made, we really appreciate the time you have spent on reviewing our manuscript.
Please find below our responses to the questions raised:
- The manuscript just described the relative configuration of partial chiral center, however the absolute configuration of all five novel compounds is not confirmed. The author should confirm the absolute configuration by ECD or single crystal diffraction.
Thank you for your remarks on structure determination of the isolated compounds. The absolute configuration of newly isolated constituents could not be assigned on the basis of NMR and MS measurements. Polyporenic acids represent a distinct group of fungal steroids with characteristic structural features identified from different polypore species. Previous studies performed on several members of this series of fungal metabolites revealed some issues related to the absolute configuration of these constituents. In a study written by King et al entitled “The stereochemistry of polyporenic acid A methyl ester” (Tetrahedron Letters, Volume 25, Issue 32, 1984, Pages 3489-3492) the absolute configuration of C25 was established based on single crystal X-ray crystallographic analysis of the methyl ester of polyporenic acid A, and it was found to be 25 (S). Another study by Kamo et al. (Anti-inflammatory Lanostane-Type Triterpene Acids from Piptoporus betulinus. J. Nat. Prod. 2003, 66, 8, 1104–1106) presents the results of a semisynthetic preparative work performed on derivatives of polyporenic acid A to determine the absolute configuration. The results confirmed that the examined fungal metabolites are (25S,3′S)-(+)-12R-hydroxy-3R-(3′-hydroxy-3′-methylglutaryloxy)-24-methyllanosta derivatives. Assuming that the same metabolic pathways lead to the recently isolated polyporenic acid N-S and the previous members of the same series of polyporenic acids, an identical configuration of chiral centers is suggested for compounds described in the current study.
- The number of the subtitle should be re-checked, especially the order of “3. Results and Discussion” and “2. Materials and Methods” (latter).
Thank you for your comment, the suggested modification was applied.
- The more details of determination of compound structure should be described, for example, in line 138, “NOE correlations of the H-3 proton”; line 153, “the NOE” and line 161, “Based on their 2D NMR correlations”
The structure determination of the compounds was further detailed in the manuscript by specifying 2D NMR correlations and adding relevant literature references.
- The table 1, please check the format of the three-wire meter.
The table has been improved according to the suggestion of the reviewer.
- Figure 1, the configuration of C-20 should be presented.
Thank you for your suggestion, the configuration of C-20 is now presented.
Reviewer 2 Report
This manuscript reported 12 polyporenic acids, including five new and seven known ones, from a mushroom Buglossoporus Q., and their efflux pump inhibitory activities on Colo 320 cells. This manuscript provide some valuable information for the readers. Overall, it is acceptable for the journal after some minor revisions as followings,
1. The absolute configurations of the new compounds should be mentioned. In most cases, this type of compounds possess the identical absolute configurations based on biosynthetic considerations. However, the absolute configurations at C-12 in compound 5 due to the substitution group of OH should be clearly assigned. At least, the NOE of H-12 should be presented in Fig. 2.
2. The NOE correlations in Fig. 2 are too simple. Please provide more key relations, such as the methyl groups.
3. Table 1: Please make it clearer for the solvent used for the NMR tests. The bottom line of table 1 are missing.
4. Section 2.3. Please include the calculated m/z, not just the deviations are showed. UV and IR could be included.
5. Please correct the format errors of references such as ref. 18, 21,and 24.
6. What are the main components in the ethyl acetate extraction? If only CHCl3 fraction was isolated, please clarify it!
Author Response
Thank you for the helpful and valuable comments you made, we really appreciate the time you have spent on reviewing our manuscript.
Please find below our responses to the questions raised:
- The absolute configurations of the new compounds should be mentioned. In most cases, this type of compounds possess the identical absolute configurations based on biosynthetic considerations. However, the absolute configurations at C-12 in compound 5 due to the substitution group of OH should be clearly assigned. At least, the NOE of H-12 should be presented in Fig. 2.
Thank you for your remarks on structure determination of the isolated compounds. Polyporenic acids represent a distinct group of fungal steroids with characteristic structural features identified from different polypore species. Previous studies performed on several members of this series of fungal metabolites revealed some issues related to the absolute configuration of these constituents. In a study written by King et al entitled “The stereochemistry of polyporenic acid A methyl ester” (Tetrahedron Letters, Volume 25, Issue 32, 1984, Pages 3489-3492) the absolute configuration of C25 was established based single crystal X-ray crystallographic analysis of the methyl ester of polyporenic acid A, and it was found to be 25 (S). Another study by Kamo et al. (Anti-inflammatory Lanostane-Type Triterpene Acids from Piptoporus betulinus. J. Nat. Prod. 2003, 66, 8, 1104–1106) presents the results of a semisynthetic preparative work performed on derivatives of polyporenic acid A to determine the absolute configuration. The results confirmed that the examined fungal metabolites (25S,3′S)-(+)-12R-hydroxy-3R-(3′-hydroxy-3′-methylglutaryloxy)-24-methyllanosta derivatives. Assuming that the same metabolic pathways lead to the recently isolated polyporenic acid N-S and the previous members of the same series of polyporenic acids, an identical configuration of chiral centers is suggested for compounds described in the current study. The determination of the position of the H-12 proton has now been confirmed in the manuscript by describing NOE correlations and citing analogies from the literature. The NOE of H-12 is now shown in Figure 2.
- The NOE correlations in Fig. 2 are too simple. Please provide more key relations, such as the methyl groups.
Thank you for your suggestion. We have now provided additional key NOE correlations; Figure 2 has been updated.
- Table 1: Please make it clearer for the solvent used for the NMR tests. The bottom line of table 1 are missing.
The table has been improved, the bottom line was added, and the font size of letters used to designate NMR solvent type was increased.
- Section 2.3. Please include the calculated m/z, not just the deviations are showed. UV and IR could be included.
Thank you for your comments. The calculated m/z values have been added. We consider that extensive NMR and MS measurements are the most suitable methods for the appropriate characterization of a certain secondary metabolite, in our view IR and UV measurements do not lead to essential scientific data regarding their structural description. Although some steroids for e.g. the insect hormone ecdysteriods do have an intense UV absorbance due to their characteristic 6-one- 7-en chromophore group in ring B of the steroid skeleton, polyporenic acid derivatives do not possess such a chromophore group therefore they do not present an intense and characteristic UV absorbance.
- Please correct the format errors of references such as ref. 18, 21, and 24
The format of references has been corrected.
- What are the main components in the ethyl acetate extraction? If only CHCl3 fraction was isolated, please clarify it!
Thank you for your question. The ethyl acetate fraction was of much lower quantity (approx. 2 g) and based on HPLC analysis did not prove to be a suitable source for isolation of polyporenic acids
Reviewer 3 Report
Comments:
This work disclosed the isolation, characterization, and potential activities of some polyporenic acids from the fruiting bodies of Buglossoporus quercinus. All compounds have been identified using 1D and 2D NMR and HRESIMS experiments. In addition, the isolated metabolites were investigated for their antiproliferative activity in vitro by MTT assay on resistant Colo 320 human colon adenocarcinoma cell line expressing P-glycoprotein (ABCB1).
In general, the purpose of the present research is clear and the experiment is carried out carefully, which can be accepted after some revision.
In abstract, IC50 should be revised as IC50; some similar description also should be checked and revised in the text.
In the part of 2.3., for the MASS analysis, the statements similar to “HRESIMS m/z 501.35693 [M+H]+” are not acceptable and should be revised. In addition, the mass value is not consistent with the test results indicated in SI.
For the structural elucidation, how did they confirm the purity of isolated metabolites? Some spectra of the metabolites contain noise signals.
Meanwhile, there are also some grammar errors and typos which they should check and summarize carefully, and all abbreviations should be defined in first use.
Comments:
This work disclosed the isolation, characterization, and potential activities of some polyporenic acids from the fruiting bodies of Buglossoporus quercinus. All compounds have been identified using 1D and 2D NMR and HRESIMS experiments. In addition, the isolated metabolites were investigated for their antiproliferative activity in vitro by MTT assay on resistant Colo 320 human colon adenocarcinoma cell line expressing P-glycoprotein (ABCB1).
In general, the purpose of the present research is clear and the experiment is carried out carefully, which can be accepted after some revision.
In abstract, IC50 should be revised as IC50; some similar description also should be checked and revised in the text.
In the part of 2.3., for the MASS analysis, the statements similar to “HRESIMS m/z 501.35693 [M+H]+” are not acceptable and should be revised. In addition, the mass value is not consistent with the test results indicated in SI.
For the structural elucidation, how did they confirm the purity of isolated metabolites? Some spectra of the metabolites contain noise signals.
Meanwhile, there are also some grammar errors and typos which they should check and summarize carefully, and all abbreviations should be defined in first use.
Author Response
Thank you for the helpful and valuable comments you made, we really appreciate the time you have spent on reviewing our manuscript.
Please find below our responses to the questions raised:
- In abstract, IC50 should be revised as IC50; some similar description also should be checked and revised in the text.
The descriptions have been corrected according to your suggestion.
- In the part of 2.3., for the MASS analysis, the statements similar to “HRESIMS m/z 501.35693 [M+H]+” are not acceptable and should be revised. In addition, the mass value is not consistent with the test results indicated in SI.
Thank you for your comment. The suggested modifications have been applied.
- For the structural elucidation, how did they confirm the purity of isolated metabolites? Some spectra of the metabolites contain noise signals.
The purity of the isolated compounds has been verified in multiple steps including chromatographic methods i.e. TLC and HPLC. Finally, the purity of compounds was also determined by 1H NMR measurements, which proved to be adequate. The signal-to-noise ratio was adequate for all samples and met the widely accepted qNMR guidelines (S/N>300).
- Meanwhile, there are also some grammar errors and typos which they should check and summarize carefully, and all abbreviations should be defined in first use.
The revised manuscript has been checked for grammar errors and typos.
Round 2
Reviewer 1 Report
The author has answered most questions.
The author has answered most questions.